# Effect of Saffron Extract on the Hepatotoxicity Induced by Copper Nanoparticles in Male Mice

**DOI:** 10.3390/molecules26103045

**Published:** 2021-05-20

**Authors:** Azza A. Attia, Heba S. Ramdan, Rasha A. Al-Eisa, Bassant O. A. Adle Fadle, Nahla S. El-Shenawy

**Affiliations:** 1Zoology Department, Faculty of Science, Alexandria University, Alexandria 21511, Egypt; azzaaattia@hotmail.com; 2Medical Research Institute, Biophysics Department, Alexandria University, Alexandria 21511, Egypt; heba@yahoo.com (H.S.R.); Bassant@gmail.com (B.O.A.A.F.); 3Department of Biology, College of Sciences, Taif University, P.O. Box 11099, Taif 21944, Saudi Arabia; stardust20117@gmail.com; 4Zoology Department, Faculty of Science, Suez Canal University, Ismailia 41522, Egypt

**Keywords:** liver, mice, copper nanoparticles, saffron, characterization, biochemical, histology, transmission electron microscope

## Abstract

Background: Nanotechnology application has widespread use in many products. Copper nanoparticles (CuNPs) are widely used in industrial applications. The present study was conducted to investigate the effect of the ethanolic saffron extract (ESE) as a natural antioxidant on the hepatotoxicity induced by CuNPs in male mice. Methods: The characterization of CuNPs was determined using ultraviolet–visible absorption spectroscopy, particle size analysis, zeta potential, Fourier-transform infrared spectroscopy, and electron microscope. The effect of saffron on the hepatotoxicity induced by CuNPs in mice was evaluated by evaluating the survival rate of the mice, oxidative stress, antioxidant capacity, DNA evaluation, as well as its effect on the histology and transmission electron microscope of the liver. Results: The results revealed that all parameters were affected in a dose-dependent manner by CuNPs. These effects have been improved when the treatment of CuNPs is combined with ethanolic saffron extract. Conclusions: We can conclude that saffron and its bioactive crocin portion can prevent CuNP-induced oxidative liver damage. This substance should be useful as a new pharmacological tool for oxidative stress prevention.

## 1. Introduction

Nanotechnology application is now in widespread use for consumer products and in new pharmaceutical, electronic, and other industries. Metal nanomaterial have become significantly used in biomedical sciences due to their interesting and unique properties [1,2]. They possess unique optoelectronic properties, due to their well-known localized surface plasmon resonance characteristics [3]. They can be synthesized and modified from their metal precursors with different chemical functional groups that allow them to be combined with antibodies, ligands, and medicines of interest, and thus open up a broad range of application spectra [4]. 

Copper nanoparticles (CuNPs) are used in industrial applications as an additive in lubricants [5], for metallic coating [6], and as a highly reactive catalyst in organic hydrogen reactions. Additionally, they have been used in osteoporosis treatment, antibacterial materials, additives in livestock, and poultry feed. It was found that these nanoparticles being released into the environment and the consequences of their various effects on human health have increased. A common mechanism of toxicity of NPs is thought to be mediated via oxidative stress [7], which damages lipids, carbohydrates, proteins, and DNA [8]. This can lead to the production of highly toxic electrophilic species as malondialdehyde (MDA) (an indicator of lipid peroxidation contents) which can affect and alter cellular signal transduction pathways [9,10]. CuNPs can affect the human body via consumer products, spillage during shipping, and handling. Excess intake exposure to it may cause hemolysis, hepatotoxicity, oxidative stress, and even renal dysfunction [11]. 

A collection of toxicological events, such as hepatocirrhosis [12], lipid profile changes, oxidative stress, renal dysfunction [11], and the activation of the mucous membrane of the food channel, can be caused by the overload of common copper in vivo. In the animal body, CuNPs induced severe impairment in the kidney, liver, and spleen in mice due to their toxicity [13,14]. The pulmonary effects of inhaled CuNPs were studied by Kim et al. [15] in mice. They observed that dystrophy or tissue necrosis was caused by the rise in its dosage to levels around its toxicity threshold. Its toxicity was associated with the size/specific surface area of the particle. The variation in the size of NPs could lead to an alteration in their catalytic properties [16]. Meng et al. [17] detected a metabolic alkalosis and copper accumulation in the kidneys in mice that were orally exposed to CuNPs. 

Doudi and Setorki [18] found that CuNPs with 10–15 nm diameters, and 99.9% purity could induce toxicity and changes in the liver and lung tissues of rats. Cholewińska et al. [19] performed a study on the effect of CuNPs on gastrointestinal and hepatic morphology and function in a rat model. CuNPs were shown to be ingested from the intestine at higher dietary doses and to accumulate in brain tissue to cause liver harm. Using antioxidant nutrients as a natural product is one of the most common forms of alternative medicines. This, because of its pharmacological and antioxidant effect [20], and its role in the prevention of oxidative stress associated with different diseases. *Crocus sativus* flower (family Iridaceae) is a flowering plant cultivated in various parts of the world, including Iran, China, Spain, Italy, and Greece [21]. It is identified as saffron, which is rich in flavonoids, vitamins, and carotenoids and consists of the dried stigmas of the plant.

*Crocus sativus* (saffron) has been used in India and other parts of the world for medicinal purposes for many decades. It has been used for nutritional and medical applications, in the food industry (flavoring and herbal agents), and in the garment industry (dyes). Furthermore, it has a golden yellow-orange color is primarily due to the presence of α-crocin. The medicinal uses of saffron can be attributed to active metabolites, such as crocin, crocetin, picrocrocin, and safranal present within it, which exhibit antihypertensive, anticonvulsant, antidepressant, antitussive, antigenotoxic, anxiolytic, aphrodisiac, antioxidant, anticancer, anti-inflammatory, antinociceptive, and muscle relaxant activity [22]. 

Moreover, there are more than 150 volatile and aroma-producing elements in saffron, many of which are carotenoids, including zeaxanthin, lycopene, and separate alpha and beta-carotenes [22]. The wide range of applications of saffron is attributed to its free radical scavenging nature, which makes it a potent chemotherapeutic agent for the treatment of various diseases. 

Saffron aqueous extract has an important role as a chemotherapeutic application in mice [23]. Patel et al. [24] explained that saffron extract exerts its anti-cancer effects via RNA and DNA synthesis inhibition, apoptosis induction [24,25,26], self-renewal genes expression reduction [27], topoisomerase inhibition [28], cell proliferation inhibition [27,28,29], immune modulation [30,31], and cell growth reduction [32,33]. Besides, Shakeri et al. [34] have reported that saffron has selective toxic and protective effects on cancer cells, has no harmful effects on normal cells and prevents the development of tumors. Safran in medicine may also be a good candidate for new drug products for the treatment of cardiovascular diseases (CVDs) [35].

Samarghandian et al. [36] showed that the daily intraperitoneal injection of ethanolic extract of saffron for 4 weeks can be effective to protect susceptible aged liver, where it ameliorated the increased serum nitric oxide (NO) and lipid peroxidation (Malondialdehyde—MDA) levels and decreased glutathione-S-transferase (GST) activity in the liver of old rats (20 months). The changes in activities of antioxidant enzymes [superoxide dismutase (SOD), GST, and catalase (CAT)] as well as the levels of MDA in the liver and serum NO determined the effect of saffron on the status of oxidative stress. Khazdair et al. [37] showed that saffron and its components can be considered as promising agents in the treatment of nervous system disorders (acts as an anticonvulsant and anti-Alzheimer property). Furthermore, saffron was suggested as a drug for diabetic neuropathy treatment. Zilaee et al. [38] studied the effects of the saffron extract on many diseases, such as allergic asthma, and the resolution of inflammation in the metabolic syndrome [39,40,41]. From the above-mentioned information, there are no data available on the effect of the saffron extract on CuNPs. Therefore, the purpose of the present study was to investigate the effect of the ethanolic saffron extract on the hepatotoxicity induced by CuNPs in male mice at biochemical, histological, and ultrastructural levels.

## 2. Results

### 2.1. Characterization of the CuNPs and Zeta Potential

The UV-visible absorption spectrum showed a typical absorption spectrum of CuNPs (Figure 1). The suspension of CuNPs has a strong absorption maximum at a wavelength of 550 nm. Particle size distribution, which was carried out by particle size analyzer showed that the CuNPs appeared with mean particle sizes of 82.87 nm. The polydispersity index was 0.5 (less than 1), indicating the homogeneous nature of the formulation (Figure 2).

The Zeta potential of the CuNPs was determined to estimate the surface charge of the nanoparticles in solution. The results revealed that the particles were negatively charged with a range of—23.9 ± 6.87 mv (Figure 3).

### 2.2. Fourier-Transform Infrared Spectroscopy (FT-I) and Morphology of CuNPs Using the Electron Microscope (EM)

The Fourier-transform infrared spectrum shows the peaks at 3426 cm^−1^, 1651 cm^−1^, and 1423 cm^−1^ (Figure 4). These peaks correspond to the hydroxyl, oxidized carbonyl group and conjugated carbonyl group, respectively, indicating the presence of the polyhydroxy structure on the surface of copper nanoparticles. The electron micrograph revealed the appearance of complete spherical-shaped particles with a low level of agglomeration, and they have a size range of 82.78 (Figure 5).

### 2.3. External Signs, Body Weight and Mortality

All experimental animals were examined daily for thirty-days to record any signs of abnormalities attributed to the two doses of CuNPs (100 and 250 mg/kg). The results showed no changes in the general health of mice, where most animals were still active. Additionally, there were no significant increases in the bodyweight of all experimental groups.

However, there was a significant decrease in the body weight of mice treated with 250 mg/kg CuNPs, compared to the control group (Table 1). The mortality was zero in the control mice group and 100 mg/kg CuNPs + 60 mg/kg of ESE as well as in those treated with 60 mg/kg of ESE. The mortality was 30% in the animals treated with 250 mg/kg CuNPs. However, only 10% of dead mice recorded in the two treated-groups with 100-CuNPs and those treated with 250-CuNPs + 60 mg/kg ESE during the experiment (Table 1).

### 2.4. Enzymes Markers in Serum of Mice

The serum enzyme markers of the liver as aspartate aminotransferase (AST), alanine aminotransferase (ALT), and alkaline phosphatase (ALP) for both control and experimental treated mice are presented in Table 2. After 30 days of administration with two doses of CuNPs, there were significant increases in the activity of AST, ALT, and ALP in the serum of treated mice, as compared to the control. There was a significant decrease in the serum activity of AST, ALT, and ALP of those treated with both doses of CuNPs with 60 mg/kg ESE as compared to those treated with 100 and 250 mg/kg CuNPs only.

### 2.5. Oxidative Stress and Total Antioxidant

The results showed marked significant increases in concentrations of malondialdehyde (MDA) as an indicator of lipid peroxidation (LPO) in the serum of mice administered with CuNPs, comparing with the control. However, there were no significant differences detected in the MDA levels of mice administered with 60 mg/kg ESE as compared to the control (Figure 6). There were significant decreases in the levels of serum MDA of those treated with both doses of CuNPs and ESE, as compared to its related dose of CuNPs groups (Figure 6).

Depending on the dose levels, the results exhibited marked significant decreases in the level of serum TAC of mice that were injected with CuNPs as compared with the control. However, there were observable increases in this parameter in the serum of mice administered with two doses of CuNPs and 60 mg/kg of ESE, as compared to those of the CuNPs groups only (Figure 7).

### 2.6. Light Microscopic Observations

The liver of control mice revealed that the hepatocytes are polygonal in shape and exhibited distinct boundaries, and they have centrally located spherical or ovoid basophilic nuclei and acidophilic cytoplasm. These nuclei showed considerable variations in size from cell to cell. Each nucleus contains one or two prominent nucleoli and dispersed chromatin (Figure 8a). The light micrographs of liver mice treated with 60 mg/kg of ESE showed that the hepatocytes had preserved their characteristic organization and became separated from each other by wide sinusoidal spaces. Their nuclei were vesicular and displaying normal appearance. The activated Kupffer cells were observed in most liver sections of the tissue (Figure 8b). 

Sections of liver mice treated with 100 mg/kg CuNPs showed that the hepatocytes were more or less similar to the control. In a few sections, they had lost their characteristic organization, and the cytoplasm was slightly vacuolated. The nuclei were vesicular and displaying their normal shaped structures. Kupffer cells appeared few and showing a marked decrease in their sizes (Figure 8c). However, the light micrographs of liver sections of mice administered with 250 mg/kg CuNPs showed severe congestion and dilatation of the blood vessels, in addition to the massive lymphocytic infiltration around them. Most of the hepatocytes were greatly damaged, exhibiting severe vacuolation, and had lost their acidophilic substances. Moreover, the nuclei in most hepatocytes were displaying pyknosis, and few small-sized Kupffer cells were observed (Figure 8d). 

Furthermore, the hepatocytes in sections of liver mice treated with 100 mg/kg CuNPs + 60 mg/kg ESE showed no evidence of damage compared to those treated only with 100 mg/kg CuNPs. However, an observable dilatation in the hepatic sinusoids was seen (Figure 8e). The light micrographs of liver mice treated with 250 mg/kg CuNPs and 60 mg/kg ESE showed that the hepatocytes normally appeared in their polygonal-shaped structure. Their nuclei were vesicular and displaying their normal shaped structures as well as the control. The Kupffer cells were normal in shape (Figure 8f).

### 2.7. Transmission Electron Microscope

Ultrathin sections of the liver of control mice revealed that the hepatocytes have a regular oval or spherical shaped nuclei with electron-dense nucleoli and outlined by a well-defined nuclear envelope (Figure 9a). The cytoplasm contains numerous densely packed mitochondria; they were rounded, elongated, and oblong. Short, flattened membranes of the rough endoplasmic reticulum (rER) and the primary lysosomes, in addition to the presence of few lipid droplets, were usually observed in the hepatocytes (Figure 9a). Furthermore, β-granules of glycogen deposits were uniformly distributed in the cytoplasm in close association with the areas of the rER. In ESE-treated mice, the hepatocytes nuclei appeared normal, where most of them showed regular nuclear envelopes with dense scattered clumps of heterochromatin. The cytoplasm of most hepatocytes contained numerous evenly distributed mitochondria. They were of different shapes and sizes, most of them were rounded, elongated, and oblong, and had distinct obvious cristae (Figure 9b).

The electron micrographs of liver mice treated with 100 mg/kg CuNPs revealed injury in the hepatocyte nuclei, and they contain large, dark, and well-developed nucleoli. In the cytoplasm, the mitochondria were rounded and elongated in shape. The rER was greatly dilated and reduced into a few short irregularly placed dilated cisternae through the cytoplasm of hepatocytes, and there were many proliferated smooth endoplasmic reticula, which indicate the toxic effect of CuNPs. Many light electron-dense lipid droplets and small dense lysosomes could be seen (Figure 9c). 

The liver section of mice treated with 250 mg/kg of CuNPs revealed the deposition of dark electron-dense spherical nanoparticles inside the nucleus of hepatocyte (Figure 9d). In the cytoplasm, the increase in glycogen contents displaces the cytoplasmic organelles to the periphery of cells, and there were many light electron density lipid droplets (Figure 9d,e). Moreover, the electron micrographs showed an apparent profile of the myelin, shown in Figure 9e.

The electron micrograph of liver mice treated with 100 mg/kg of CuNPs and 60 mg/kg ESE revealed an improvement in the structures of hepatocytes. Their nuclei were regular in shape and had electron-dense nucleoli. The mitochondria in the cytoplasm were evenly distributed with densely packed structures. The rER was frequently evaluated when short-flattened membranes were present near the nucleus, and if the cytoplasm contained many lipid droplets and many dense lysosomal bodies (Figure 9f).

The hepatocytes of liver mice treated with 250 mg/kg of CuNPs and 60 mg/kg ESE showed little improvement in the structure of the nucleus, compared to those of the treated group with 250 mg/kg CuNPs. In the cytoplasm, the mitochondria were characterized by the marked disappearance of their cristae, and the profiles of rER were interrupted (Figure 9g).

DNA analysis with the agarose gel electrophoresis, the results showed massive DNA fragmentation, which appears as DNA laddering (smearing) in the liver tissues of mice administered with two doses of CuNPs. However, partial protection against apoptotic and necrotic cells appeared in groups treated with 100 and 250 mg/kg CuNPs and 60 mg/kg ESE, which showed low DNA laddering (DNA fragmentation). Furthermore, no detectable changes were observed in the liver tissue of ESE and control mice (Figure 10). The DNA intact band appears to be condensed near the application point with no DNA smearing.

## 3. Discussion

In the present study, the toxicity of intraperitoneal injection with CuNPs (100 and 250 mg/Kg body weight) for 30 days on the liver of mice was evaluated. These doses do not necessarily reflect the actual concentrations of CuNPs found in real environments. However, they can be used to assess the health risks from exposure to CuNPs.

Concerning the characterization of CuNPs, the present results showed that the crystalline structure morphology of synthesized CuNPs was observed using their optical properties characterized by UV–visible spectroscopy, FT-IR analysis, and TEM. The results revealed that the UV–Vis absorption of the prepared CuNPs in this study showed a typical spectrum. With the TEM, the results showed that CuNPs were spherical, having a smooth surface and the size of the particles was 82.87 nm. 

FT-IR is an effective method to reveal the functional groups of the prepared CuNPs, at peaks 3426/cm, 1651/cm, and 1423/cm. These peaks correspond to the hydroxyl, oxidized carbonyl group, and conjugated carbonyl groups, respectively. These results are similar to the results of Xiong et al. [42] who found that the CuNPs indicated the presence of a polyhydroxy structure on the surface of CuNPs. 

The current study revealed that the rate of mortality was increased in mice administered with a high dose of CuNPs (250 mg/kg), and there were significant decreases in the body weights of this group of mice as compared to the control animals. This suggests that exposure exerts a major visible impairment in the health statuses of animals. However, no significant differences in body weight were observed in mice administered with a low dose of CuNPs, as compared to the control. Lei et al. [14] found that incidences of toxicity were higher in a higher dose of CuNPs (250 mg/kg) compared to the low dose (50 mg/kg), which parallels the present study. 

Depending on the dose of CuNPs, the current results revealed a significant elevation in the activity of serum AST, ALT, and ALP of mice administered with 250 mg/kg CuNPs. Serum aminotransferases analyses have become a standard measure of hepatotoxicity because of the significance of these enzymes. AST and ALT are located inside the cell, while the ALP is located on the cell membrane. These transaminases play an important role in protein and amino acid metabolism. In such cases, the serum AST rises and escapes from the damaged hepatic cells into the serum. These enzymes are usually found in the liver and other tissues where they act in the metabolism of energy requiring amino acid transamination. However, in cases of cellular damage, the AST and ALT could leak out into the general circulation, leading to elevated activity [43]. These changes in the levels of these enzymes are in association with a pathology, involving the necrosis of the liver. Gaskill et al. [44] had reported that the release of ALT, AST, and ALP from the cytoplasm of hepatic cells could have occurred secondary to cellular necrosis. This may be due to the leakage of these enzymes from the cytoplasm of the liver cells into the general circulation, liver dysfunction, and disturbance in the biosynthesis of these enzymes, with an alteration in the permeability of the liver membrane. 

The obtained results are consistent with the previous findings of Doudi and Setorki [19], who reported that CuNPs were found to increase in these three enzymes AST, ALT, and ALP. Chen et al. [13] had reported that increase ALP reflected hepatic dysfunction. Additionally, the increase in ALP was consistent with the liver toxicity induced by AgNPs, [45,46]. Mansouri et al. [47] suggested that, due to the damaged to the liver cells, such enzymes are released into the blood. Therefore, a high amount of these enzymes indicates the destruction of liver cells. Furthermore, the toxic effect of nanoparticles can be elucidated by the estimation of oxidative stress parameters [48]. Nanoparticles are found to produce free radicals and to induce oxidative stress and thus can disturb the antioxidant defense system in the animal [49]. 

The current results revealed increased levels of MDA and decreased the TAC in the serum of mice administered with 100 and 250 mg/Kg, depending on the dose level. These results follow the results of Al-Salmi et al. [50], who found that CuNPs caused an increase in the level of MDA. Grotto et al. [51] explained that MDA is one of the end products of lipid peroxidation, and the production of it is used as a biomarker to measure the level of oxidative stress in an organism. Free-radical-induced LPO has been suggested to alter the membrane structure and function, thus causing cellular abnormalities, such as mutations and cell death. LPO could produce a gradual loss of cell membrane fluidity, reduce membrane potential, and increase permeability to ions such as Ca^2+^ [52]. 

Knaapen et al. [53] suggested that free radicals also interact with lipids and proteins, abundantly present in bio-membranes, to yield the LPO products associated with mutagenesis and apoptosis. It is well established that the uncontrolled generation of ROS triggers a cascade of proinflammatory cytokines and mediators via activation of and NF-κB signaling pathway that controls the transcription of inflammatory genes, such as IL-1β, IL-8, and TNF-α [54]. The NF-κB group of proteins activates genes responsible for defense mechanisms against cellular stress and regulates miscellaneous functions, such as inflammation, immune response, apoptosis, and cell proliferation.

The studies of Fahmy and Cormier [55] indicated that, in comparison with normal cells and cells exposed to CuNPs, the level of antioxidant enzymes (CAT and glutathione reductase activity) decreased, and the activity of glutathione peroxidase was increased. They suggested that the increase in the activity of GSH is due to that CuNPs not only produce free radical but they also stop the cell antioxidant defense.

Mansouri et al. [47] had reported that ZnNPs caused a significant increase in the hepatic level of MDA, indicating the presence of oxidative stress. The current results are confirmed by the study of Ranjbar et al. [56], who reported that silver NPs, after 14 days of exposure, caused a significant decrease in TAC. These data suggest that the NPs can induce oxidative damage through an ROS-mediated process However, it remains to be investigated whether NPs directly or indirectly induce free radicals through depletion of antioxidant defense mechanisms caused by interactions with antioxidant systems [57]. Furthermore, in the present study, the intraperitoneal administration of ESE in mice for 30 days showed an improvement in the biochemical and histological alterations induced in the livers of mice by CuNPs, depending on the dose levels. Additionally, our data were supported by the results and observations of the liver tissues. Similarly, Anlin et al. [58] reported that ESE has a curative effect against liver toxicity induced by alcohol and CCl_4_ in rats. 

The present results revealed a marked and significant decrease in serum levels of AST, ALT, and ALP in mice administered with the ESE. These results explained the role of saffron in improving the disruption in the liver function enzymes and its role in alleviating the increased levels of LPO. Additionally, Mohajeri et al. [59] had reported that the ESE ameliorated rifampin-induced histopathological and biochemical alterations ALT, AST, and ALP in rats. Additionally, Mohajeri and Doustar [60] found that the ESE could reduce LPO and improve antioxidant enzyme activity—SOD-, CAT-, and GSH-related enzymes—in the liver of rats, the liver being the target organ for CuNPs and the site at which they preferentially accumulate [61].

Iranshahi et al. [62] showed that ESE (stigma and petal) reduces the incidence of hepatic necrosis in the range of 29.2% to 93.9% induced by CCl_4_ in mice. Additionally, these findings confirm other findings showing the protective effects of ESE on hepatic tissue ischemia-reperfusion-induced oxidative damage in rats [63]. Yosry and Seham [64] suggested that the mechanism of ESE improved hepatotoxicity through the inhibition of oxidative stress and enhancing the antioxidant defense system. Zheng et al. [65] reported that the carotenoids in saffron extracts may protect tissues from oxidative damages due to their antioxidant effect. Additionally, they showed remarkable modulation in the levels of oxidative markers in the liver. 

In the current study, the cytotoxicity of CuNPs was evidenced by the appearance of morphological changes in liver cells, such as congestion and dilatation of the blood vessels, the presence of much inflammatory infiltration in pre-central veins, as well as the appearance of hepatocellular necrosis. These alterations were prominent in the liver tissues of mice administered with 250 mg/kg of CuNPs. However, the decrease in these degenerative changes in the liver of mice was apparent in the liver tissues of mice treated with ESE. Our present study, parallel with Kim et al. [46], reported some histopathological changes and increased infiltration around the central vein induced by nanoparticles. The destruction of lobular structure and vacuolization of hepatocytes together with the dilatation of the central vein and blood sinusoids indicate that these NPs may affect the permeability of the cell membrane in hepatocytes and the endothelial lining of blood vessels [66]. Additionally, the appearance of inflammatory cells in hepatic tissue suggests that nanoparticles can interact with proteins and enzymes in the interstitial tissue of the liver, interfering with the antioxidant defense mechanism and leading to the generation of ROS, which in turn may induce an inflammatory response [67]. 

The obtained results are consistent with the previous findings of Doudi and Setorki [19] who found that 100 mg/kg CuNPs induced histological changes in the hepatic tissues, vasculature in the central vein, and the disappearance of hexagonal liver lobules which is lined with the present study. Moreover, the destruction of the lobular structure, vacuolization of hepatocytes (fat deposits), and congestion of RBC, and infiltration of leukocytes in the liver tissue of mice administered with 300 mg/kg ZnNPs, indicating its necrotic effects [56]. 

Using the TEM, the results revealed many ultrastructural abnormalities in the liver tissues of mice administered with 250 mg/kg CuNPs. Most nuclei of the hepatocytes were pyknotic with condensed chromatin. Besides, they contained more than two large, dark, and well-developed nucleoli. These indicate the hypermetabolic activity of the liver, cellular and nuclear degeneration, cellular atrophy or rupture, indicated by the direct toxicities of CuNPs. Moreover, the pyknotic nuclei refer to an irreversibly condensed form of chromatin material in the nucleus, which indicated highly severe and irreversible damage to the liver [68]. This change in the nucleus indicates that this organelle is affected in a major way by CuNPs. Colegio et al. [69] and Waisberg et al. [70] suggested that chromatin condensation was due to the progressive inactivation of the nuclear component, probably due to the inhibition of DNA repair and DNA methylation.

Furthermore, liver toxicity seen in the current study may be explained by the deposition of CuNPs in the nuclei of certain hepatocytes, leading to the generation of ROS associated with inflammatory, oxidative stress, genotoxicity, and the induction of apoptosis. It had been reported that some NPs, owing to their small size, are capable of reaching the nucleus and might attack DNA [71]. They may also exhibit an indirect effect on DNA through their ability to generate ROS. Lopotko et al. [72] have demonstrated that the interaction of nanoparticles with cells leads to the formation of NP clusters in various cellular organelles viz., endosomes, and vacuoles via endocytosis.

The presence of NPs inside the cell nucleus and lysosomes indicates the important role in the apoptotic process: the lysosomal proteases can be released into the cytosol, where they cause pro-apoptotic proteins to adopt their conformation and to insert into mitochondrial membranes and induce the mitochondria-mediated apoptosis pathway [73,74].

In the cytoplasm, the results revealed the appearance of fatty deposits, which may be due to LPO, which leads to rough endoplasmic damage and the detachment of the cytoplasmic lipoprotein, which indicates abnormal fat metabolism. The mitochondria showed a marked disappearance of the mitochondrial cristae. This may reflect the disturbances in ox reduction processes taking place in the organelle [75]. Mitochondria represent the most active cellular redox organelles and the localization of these particles into such redox-active centers is expected to cause alterations in various antioxidant enzyme systems that are functional here. Thus, as it happens, in the case of nanoparticles of various chemistries, particles are taken up into cells via endocytic pathways and are mainly localized to mitochondria [76,77,78]. 

Additionally, there was an increased proliferation in the number of secondary lysosomes in the cytoplasm of many hepatocytes of the liver of mice administered with 250 mg/kg CuNPs. This indicates an attempt to digest these NPs, and this is considered as a general manifestation of injury. The sequestration of damaged organelles in lysosomes is a mechanism of cellular repair and follows all types of sublethal injury [79]. 

Furthermore, the Kupffer cells were found as an indication of the damaging effects of CuNPs in the liver tissues. These cells appeared with marked irregularities in their nuclear counter and having many dense patches of heterochromatin. This indicates their activation and suggests hepatocellular damage. Kupffer cells play a defensive role, protecting against bacterial or viral invasion into the portal blood flow [80]. Other investigators had reported that Kupffer cells stimulate the DNA synthesis of hepatocytes by producing and releasing certain factors, such as prostaglandin E_2_ [81,82]. 

In the present results, DNA laddering electrophoresis was chosen for the assessment of DNA damage on the liver tissue of mice treated with different doses of CuNPs and/with the ESE. Depending on the dose levels of CuNPs, the present results showed that massive DNA fragmentation was found in the liver tissues of mice. Fahmy and Cormier [55] explained that nano copper can trigger both intrinsic and extrinsic apoptotic pathways in oxidative stress. Apoptosis has been implicated as a major mechanism of cell death caused by NPs-induced oxidative stress [83]. 

Apoptosis is associated with a distinct set of biochemical and physical changes involving the cytoplasm, nucleus, and plasma membrane. Early in apoptosis, the cells round up, losing contact with their neighbors, and shrunk. In the cytoplasm, the endoplasmic reticulum dilates and the cisternae swell to form vesicles and vacuoles. In the nucleus, chromatin condenses and aggregates into dense compact masses [84]. Martinez et al. [85] explained that ROS is known to react with DNA molecules, causing damage to both purine and pyrimidine bases, as well as the DNA backbone. ROS attack on DNA generates a huge range of base and sugar modification products. 

Among the different apoptotic pathways, the intrinsic mitochondrial apoptotic pathway plays a major role in metal NP-induced cell death since mitochondria are one of the major target organelles for NP-induced oxidative stress and apoptosis [86]. Jeng and Swanson [87] explained that metals can generate free radicals via the Fenton-type reactions that react with cellular macromolecules and induce oxidative stress.

Furthermore, the administration of saffron resulted in reduced DNA fragmentation which was confirmed by Premkumar et al. [88]. The decrease in DNA fragmentation in treatment with Saffron was similar to the results of Yosry and Sehm [64] using different techniques for the Feulgen reaction, improving the amount of DNA-containing particles. A possible mechanism of toxicity is proposed, which involves the disruption of the mitochondrial respiratory chain by CuNPs, leading to the production of ROS and the interruption of ATP synthesis, which in turn causes DNA damage. This leads to cell-cycle arrest in the G_2_/M phase. 

## 4. Materials and Methods

### 4.1. Chemicals

The copper chloride dehydrates (CuCl_2_·2H_2_O-97%) and L-ascorbic acid (vitamin C-98%) were purchased from Sigma-Aldrich, St. Louis, MO, USA.

### 4.2. Synthesis of Copper Nanoparticles

In a synthetic procedure, copper-nanoparticles were obtained via a wet chemical reduction route. Copper chloride (CuCl_2_) aqueous solution was prepared by dissolving 0.02 M CuCl_2_ in 50 mL de-ionized water in a flask. Then, the solution was heated to 80 °C in an oil bath with continuous magnetic stirring. L-ascorbic acid aqueous solution (50 mL of 0.1 M) was added dropwise while stirring. Over time, the color of dispersion was gradually changed from white to yellow to orange to brown and finally to dark brown. The appearance of yellow color followed by orange color indicated the formation of fine nano-scale copper particles from L-ascorbic acid-assisted reduction. The obtained result was centrifuged for 15 min [80]. The doses of nano-copper (CuNPs) in this preparation were 100 and 250 mg/kg, according to Lei et al. [14].

The supernatant of the prepared nanoparticles was frozen by vacuum using a freeze-drying machine (Telestar, Spain) (Model/CRYODOS-50) at City of Scientific Research and Technological Applications, Borg Al-Arab, Alexandria, Egypt (it works at 230 V, 50 Hz). The obtained solutions were placed in 50 mL falcon tubes, frozen in liquid nitrogen and frozen drying by vacuum Freeze Drying Machine at a pressure of 26.5 pa and 5.0% saccharine that was used as a cytoprotectant. The prepared CuNPs were suspended in saline and dispersed by ultrasonic vibration (130 W, 20 kHz) for 20 min for all experimental work.

### 4.3. Preparation of Ethanolic Saffron Extract

One hundred grams of saffron stigmas were macerated and shacked for 3 days in one liter of 80% ethanol at room temperature by using an Environ Shaker (Model/Lab-Line 3527 Orbit Environ Shaker, Southwest Science, P.O. Box 144, Roebling, NJ 08554, USA). Then, it was filtered, and the filtrate was concentrated under vacuum using the rotatory evaporator (40 °C) till giving dark red residues kentron by using a rotatory evaporator (model/strike 201), at Medical Technology Center, Alexandria University. The obtained ethanol saffron extract (ESE) was chilled in the refrigerator until use [89].

### 4.4. Animal Design and Care

Sixty adult male albino mice (25–29 g) were obtained from the animal house of Medical Research Institute, Alexandria University, Egypt. They were housed (5 mice/cage) in the stainless-steel cages, and they were observed for health status and acclimatized for two weeks before use at 25 ± 2 °C and humidity 55 ± 5% using normal photoperiod day/night. Mice were allowed free access to food and drinking water throughout the study. Mice were cared for following the guidelines of the Care and Use of Laboratory Animals committee, Alexandria University, Egypt. 

After two weeks of acclimatization, mice were assigned to six groups (10 mice/each) given day after day for 30 days the following treatment. The mice have received 0.5 mL saline solution and presented as group I (control). Group II was injected i.p. with 60 mg/kg ESE. Mice were injected i.p. with 100 or 250 mg/kg CuNPs, respectively, and considered as group III and IV. Groups V and VI were injected i.p. with 100 or 250 mg/kg CuNPs and followed by 60 mg/kg of ESE after each other.

### 4.5. Toxicological Assessment of CuNPs

The mice were inspected for clinical signs of toxicity by testing their emotion as excitability and aggressiveness. The autonomous functions such as diarrhea, dieresis, salivation, and mortality were evaluated during the time of dosing of 30 days. 

### 4.6. Characterization of CuNPs

#### 4.6.1. Ultraviolet–Visible Absorption Spectroscopy (UV-Vis)

UV–vis spectroscopy was used to characterize the optical absorption properties of copper nanoparticles. The absorption spectra of samples were recorded in the wavelength range of 250 to 650 nm using a UV-6800 UV\VIS spectrophotometer (JENWAY, Bruker, Ettlingen, Germany), at the Medical Biophysics Department, Medical Research Institute, Alexandria University. CuNPs suspension (1 mL) was placed in a quartz cuvette to record its wavelength absorption.

#### 4.6.2. Particle Size Analysis and Zeta Potential

The particle size of the prepared particles was determined by Laser Light Scattering on a Nano Zeta sizer particle analyzer (Malvern, UK). The laser obscuration range was maintained at 15–20%, and the mean particle size was measured after experimenting with triplicate. Using cumulative analysis software and the exponential sampling method, hydrodynamic size distribution, and surface charge (zeta potential) were determined.

#### 4.6.3. Fourier-Transform Infrared Spectroscopy (FT- IR)

Fourier-Transform Infrared Spectra of the prepared particles were obtained on a Shimadzu FTIR-8400S (Tokyo, Japan). The pellets were prepared by grinding 4–8 mg of samples with 200 mg potassium bromide. Tablets were prepared then fixed on the holder to be examined. The spectra were scanned over the wavenumber range of 4400 to 350/cm.

#### 4.6.4. Transmission Electron Microscope (TEM) for Nanoparticles

The size and shape of the prepared nanoparticles were determined by transmission electron microscope (Jeol, JSM-6360LA, Joel, Japan). For this purpose, the particle suspension was diluted 10-times with distilled water, and a drop of an aqueous suspension containing the biosynthesized CuNPs was placed on 400-mesh copper grids coated with carbon film after allowing the water to evaporate.

### 4.7. Body Weight and Biochemical Parameters

In one-week intervals over 30 days, all mice were weighed at the end of the experiment, they were euthanized using diethyl ether, and the blood samples were collected by a syringe from the inferior vena cava by heart puncture in EDTA tubes. 

Serum samples were obtained by centrifugation of the blood samples at 3000 rpm, for 30 min, and were frozen at −20 °C until assayed. The serum levels of malondialdehyde (MDA) [90], total antioxidant capacity (TAC) [91], aspartate, and alanine aminotransferase (AST and ALT) [92], as well as alkaline phosphatase (ALP) [93], were determined.

### 4.8. Qualitative DNA Fragmentation Assay Using Agarose Gel Electrophoresis

The liver tissues were excised out and preserved at −80 °C for DNA analysis. DNA fragments isolated from cells or tissue that were laddering were determined on 1.2% agarose (15 μg/lane), and were illuminated with 300 nm UV light, and a photographic record was made [94].

### 4.9. Light Microscopic Investigation

Small pieces of the liver tissues were excised carefully, fixed in 10% formalin solution, dehydrated using ascending graded series of ethanol, and cleared by xylem. The specimens were embedded in paraffin wax according to the routine processing protocol of Bancroft and Gamble [95]. Serial sections at 5 µm slices were cut and stained with hematoxylin and eosin (H&E), examined, and photographed under light microscopy.

### 4.10. Transmission Electron Microscopic Study

The small pieces of liver tissue were immediately fixed in 4% formalin and 1% glutaraldehyde, rinsed in 0.1 M phosphate buffer (pH 7.4) at 4 °C for 24 h, and post-fixed using 1% buffered osmium tetroxide (OsO_4_) at 4 °C for 2 h. Then, the specimens were washed several times with phosphate buffer for 30 min, dehydrated through ascending grades of ethanol concentrations, and at 4 °C they were treated with propylene oxide and embedded in a mixture of 1:1 of Epon-Araldite resin mixture [96]. Thick ultrathin sections (60 nm) were mounted on 200 mesh naked copper grids, double stained with uranyl acetate and lead citrate for 30 min, and lead citrate for 20–30 min, examined and photographed at Jeol Transmission Electron microscope (JSM-6360LA, Joel, Japan).

### 4.11. Statistical Analysis

Data were expressed as mean values ± SE, and statistical analysis was performed using one-way ANOVA to assess significant differences among treatment groups. Statistical significance was considered at *p* < 0.05. All statistical analyses were carried out using SPSS statistics version 16 software package (Philadelphia, PA, USA).

## 5. Conclusions

In conclusion, the present work demonstrates that saffron and its bioactive component, crocin, can prevent oxidative damage of the liver induced by CuNPs. Thus, these substances should be useful as new pharmacological tools for alleviating oxidative stress.

## Figures and Tables

**Figure 1 molecules-26-03045-f001:**
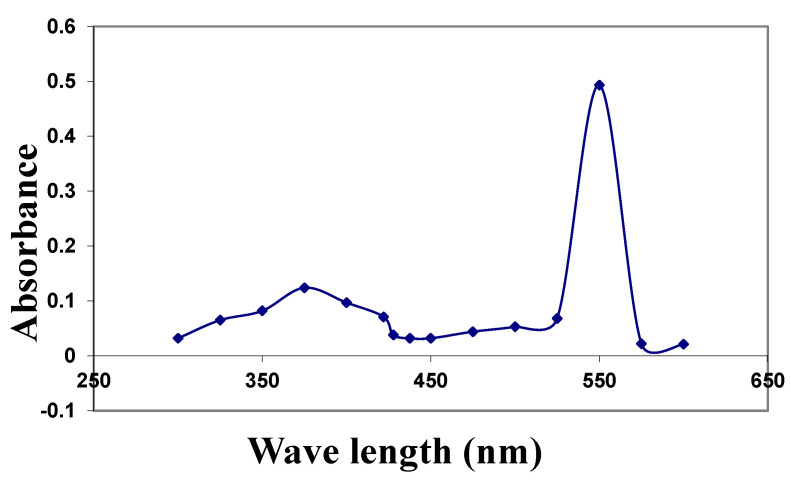
The absorption spectra of prepared CuNPs (λmax = 550 nm).

**Figure 2 molecules-26-03045-f002:**
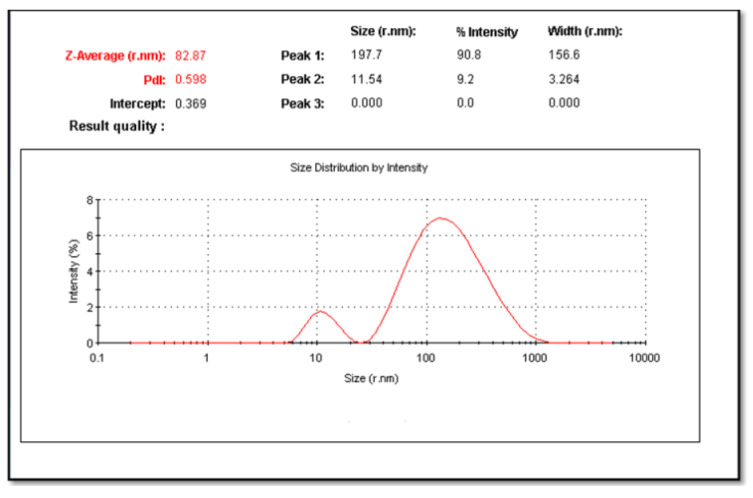
The particle size distribution curve of prepared CuNPs.

**Figure 3 molecules-26-03045-f003:**
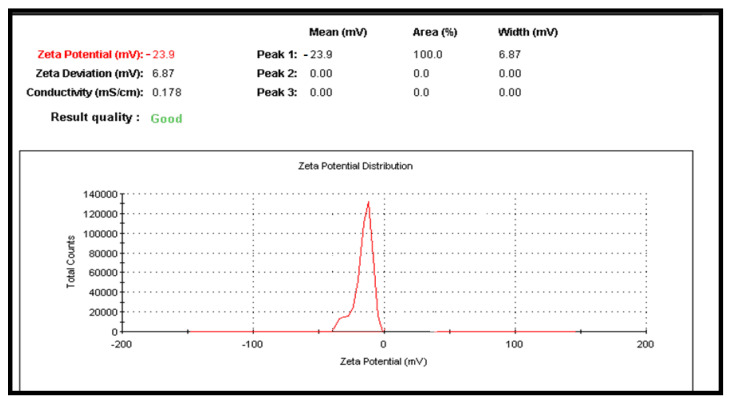
Showing the Zeta Potential curve of prepared CuNPs.

**Figure 4 molecules-26-03045-f004:**
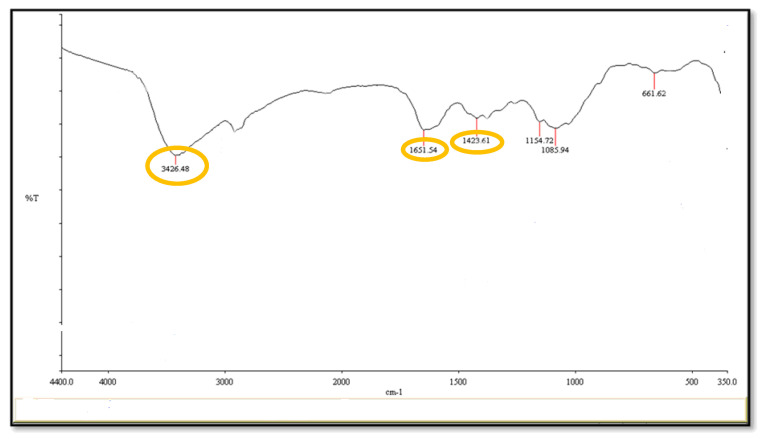
Fourier-transform infrared spectrum showing the functional groups of prepared CuNPs.

**Figure 5 molecules-26-03045-f005:**
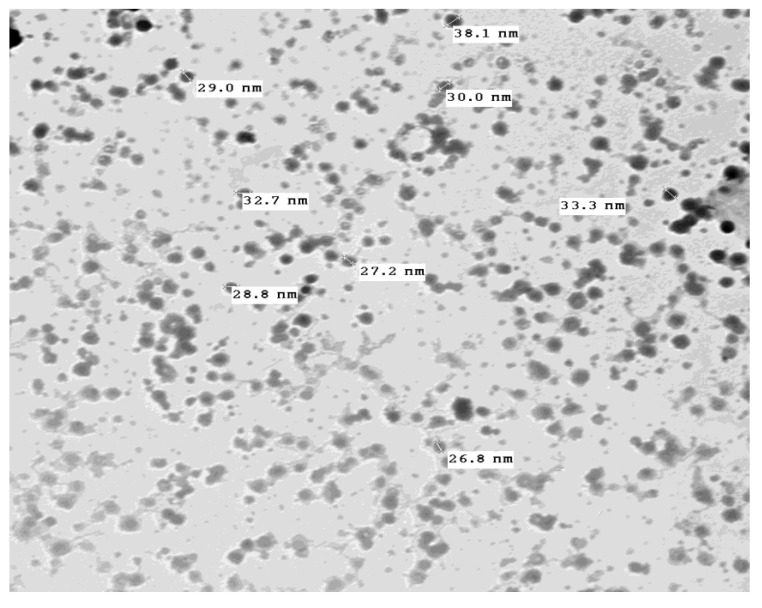
Electron micrograph showing spherical shaped particles of 82.87 nm in diameter.

**Figure 6 molecules-26-03045-f006:**
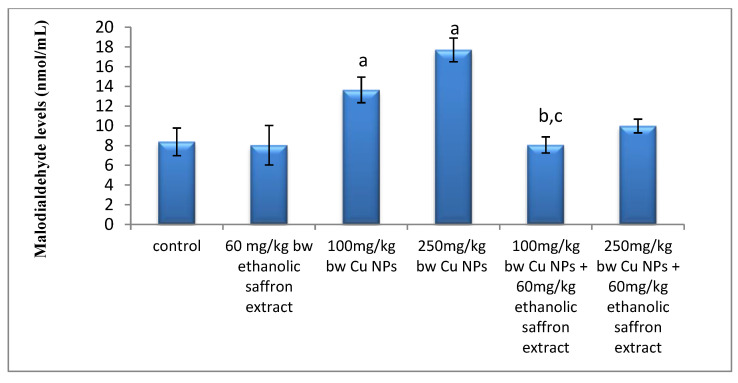
The effect of the ethanolic saffron extract on malondialdehyde levels of mice. Data are presented as mean ± SE. ^a^ Significance was considered, compared with the control group (*p* ≤ 0.05), ^b,c^ significance was considered compared with 100 and 250 mg/kg of CuNPs groups, respectively.

**Figure 7 molecules-26-03045-f007:**
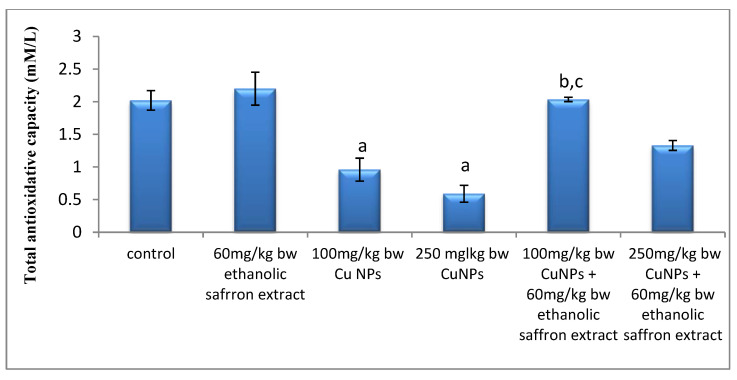
The effect of the ethanolic saffron extract on the serum total antioxidative capacity of mice. Data are presented as mean ± SE. ^a^ Significance was considered compared with the control group (*p* ≤ 0.05), ^b,c^ significance was considered compared with 100 and 250 mg/kg of CuNPs groups, respectively.

**Figure 8 molecules-26-03045-f008:**
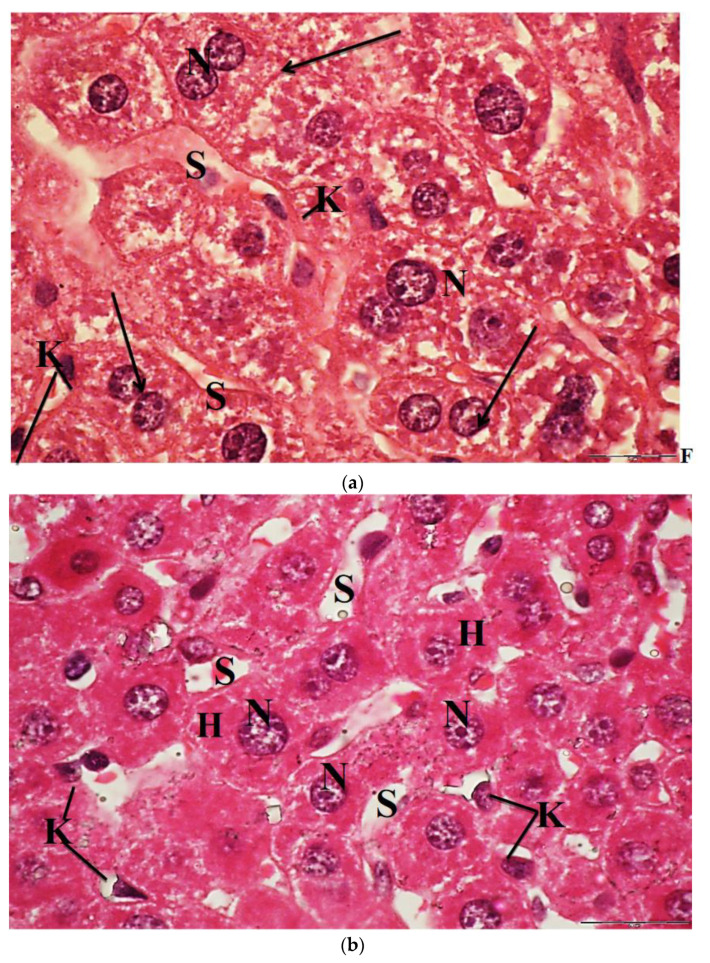
(**a**) Light micrograph of mice liver of control group showing, the vesicular spherical nuclei (N), binucleated hepatocytes (arrow); hepatic sinusoids (S); triangular-shaped Kupffer cells (K), H&E, ×1000. (**b**) Light micrograph of liver mice treated with 60 mg/kg bw of ethanolic saffron extract, showing polygonal hepatocytes (H); vesicular nuclei (N); wide blood sinusoids (S); many activated Kupffer cells (K), H&E, ×1000. (**c**) Light micrograph of liver mice treated with 100 mg/kg bw of CuNPs showing vacuolated hepatocytes; nuclei (N); few leucocyte infiltrations (thick arrow) and small size Kupffer cells (K), H&E. ×1000. (**d**) Light micrograph of liver mice treated with 250 mg/kg bw of CuNPs showing degeneration and vacuolated hepatocytes; pyknotic nuclei (arrowheads); Kupffer cells (K), H&E, ×1000. (**e**) Light micrograph of liver mice treated with 100 mg/kg bw of CuNPs + 60 mg/kg bw ethanolic saffron extract showing polygonal-shaped hepatocytes containing vesicular nuclei (N). Note: dilated blood sinusoids (S) and few small-sized appearances of Kupffer cells (K), H&E, ×1000. (**f**) The light micrographs of liver mice treated with 250 mg/kg bw of CuNPs + 60 mg/kg bw ethanolic saffron extract showed vacuolated hepatocytes (H) with vesicular nuclei (N) and Kupffer cells (K), H&E, ×1000.

**Figure 9 molecules-26-03045-f009:**
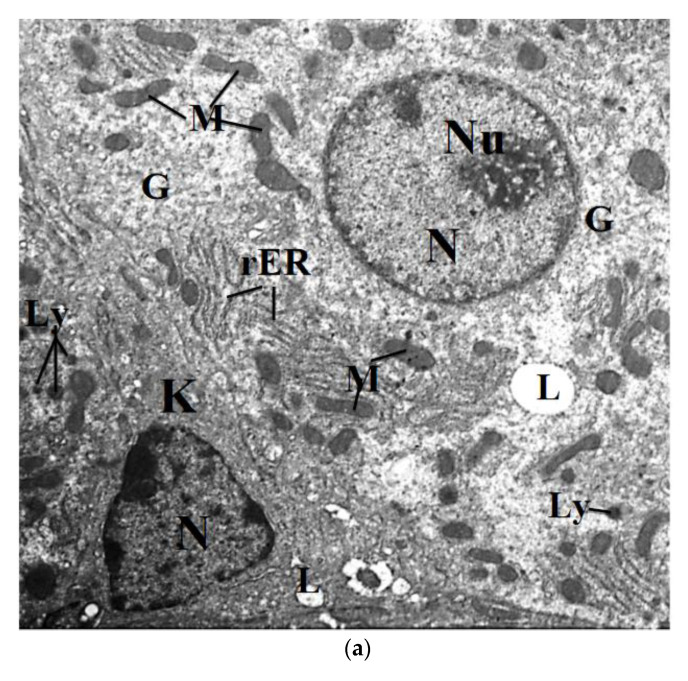
(**a**) An electron micrograph of a control mouse liver showing hepatocyte with regular nucleus (N), nucleolus (Nu); elongated shaped mitochondria (M); short profile rough endoplasmic reticulum (rER); lipid droplets(L); Kupffer cell (K) with triangular-shaped nucleus (N), ×2500. (**b**) Electron micrograph of liver mice treated with 60 mg/kg bw ethanolic saffron extract showing, the vesicular nucleus (N) of a hepatocyte, containing large nucleolus (Nu). Note that the cytoplasm contains numerous distributed mitochondria (M) and glycogen granules (G), ×2500. (**c**) Electron micrograph in liver mice administrated with 100 mg/kg of CuNP showing, hepatocytes with irregular nucleus (N), dark nucleoli (Nu), proliferated smooth endoplasmic reticulum (sER), many lipid droplets (L), many small dense lysosomes (Ly) and increase in glycogen content (G), ×2500. (**d**) Electron micrograph in liver mice administrated with 250 mg/kg of CuNPs, showing necrotic hepatic tissue and cytoplasm devoid of most organelles; aggregation of mitochondria (M) and deposition of CuNPs in nucleus (arrow), ×2000. (**e**) Electron micrograph in liver mice administrated with 250 mg/kg of CuNPs showing necrotic hepatic tissues and cytoplasm contains many lipid droplets (L); aggregation of mitochondria (M) and myelin figures (thick arrows), ×4000. (**f**) Electron micrograph of the liver mice treated with 100 mg/kg bw of CuNPs + 60 mg/kg ethanolic saffron extract showing, hepatocyte contains elongated-shaped mitochondria (M); short profile rough endoplasmic reticulum (rER); many dense lysosomes and lipid droplets (L), 3000×. (**g**) Electron micrograph of the liver mice treated with 250 mg/kg bw CuNPs + 60 mg/kg bw ethanolic saffron extract showing hepatocyte with regular nucleus (N); elongated shaped mitochondria (M); dilated rough endoplasmic reticulum (rER); lipid droplets (L); glycogen granules (G), ×2500.

**Figure 10 molecules-26-03045-f010:**
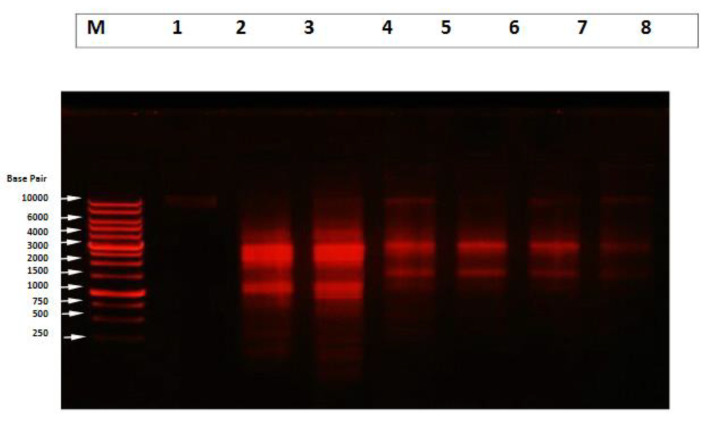
DNA analysis in the liver tissue of mice, demonstrating apoptotic and necrotic cell death induced by CuNPs and protection by ESE. Lane 1: control. Lane 2: Low CuNPs treated mice. Lane 3: High CuNPs treated mice. Lanes 4 and 5: High CuNPs, and ESE treated mice. Lanes 6 and 7: Low CuNPs and ESE treated mice. Lane 8: saffron.

**Table 1 molecules-26-03045-t001:** Percentage of mortality and body weight change of all experimental mice for 30 days.

Experimental Groups	No. of Mice	No. of Dead Mice	% of Mortality	Mean of Body Weight (g)		The Mean Change in Body Weight
				Initial	Final	
Control	10	0	-	22.3 ± 2.081	28.34 ± 1.15	27.09 ± 7.16
60 mg/kg bw ethanolic saffron extract	10	0	-	24.33 ± 1.15	30.6 ± 1.54	26.02 ± 2.06
100 mg/kg bw CuNPs	10	1	10	23.01 ± 1.32	28.66 ± 1.5	24.78 ± 7.2
250 mg/kg CuNPs	10	3	30	22.66 ± 2.5	26.66 ± 1.5	18.24 ± 9.02 ^a^
100 mg/kg bw CuNPs + 60 mg/kg ethanolic saffron extract	10	0	-	23.67 ± 1.15	29.66 ± 0.57	25.62 ± 8.33
250 mg/kg bw CuNPs + 60 mg/kg ethanolic saffron extract	10	1	10	22.66 ± 2.3	28 ± 1.73	23.88 ± 5.2

Data presented as means ± SE. ^a^ Significant difference as compared to the control group (*p* ≤ 0.05).

**Table 2 molecules-26-03045-t002:** Effect of ethanolic saffron extract on serum enzymes activity of liver functions of induced mice with CuNP.

	Experimental Animals					
Enzymes Activity (U/mL)	Control	60 mg/kg ESE	100 mg/kg bw CuNP	250 mg/kg bw CuNP	100 mg/kg CuNP + 60 mg/kg bw ESE	250 mg/kg CuNP + 60 mg/kg bw ESE
AST	60.18 ± 2.03	50.3 ± 2.4	68.84 ± 5.6 ^a^	80.55 ± 4.6 ^a^	51.66 ± 4.3 ^b^	53.35 ± 1.8 ^c^
ALT	68.44 ± 12.9	60.81 ± 5.1	79.08 ± 2.2 ^a^	90.38 ± 9.01 ^a^	60.3 ± 5.80 ^b^	61.95 ± 7.2 ^c^
ALP	134.46 ± 2.4	122.52 ± 2.07	139.56 ± 15.7 ^a^	146.6 ± 0.4 ^a^	120.56 ± 3.7 ^b^	128.17 ± 8.3 ^c^

Data are presented as mean ± SE. AST; aspartate aminotransferase, ALT; alanine aminotransferase, ALP; alkaline phosphatase, ESE; ethanolic saffron extract, and CuNP; copper nanoparticle. ^a^ Significance was considered, compared with the control group (*p* ≤ 0.05), ^b,c^ significance was considered compared with 100 and 250 mg/kg bw CuNPs groups, respectively.

## Data Availability

The data presented in this study are available on request from the corresponding author.

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
