# Peer review of "Effect of Saffron Extract on the Hepatotoxicity Induced by Copper Nanoparticles in Male Mice"

_molecules, 2021, doi:10.3390/molecules26103045_

Round 1

Reviewer 1 Report

  1. The authors explored the topic and they obtained the purpose of the study. They showed that ethanolic saffron extract can prevent CuNP-induced oxidative liver damage.
  2. The paper is well written and text is clear to read.
  3. The methods used are sufficiently documented and allow replication studies. Results obtained are well explained and data interpretation is also correct. Conclusions are consistent with the evidence and arguments presented.
  4. About limitations the authors should modify and standardize statistical analysis.

Specific indications:

  1. The following acronym should be reported in extenso at their first appearence in the text: line 88 CVDs;
  2.  In the legend of figures and tables the authors should add informations about *P values and they should make it clear in the figures by placing asterisk/asterisks above each treatment group if differences with the reference group are significant;
  3. In addition statistical analysis should be revised;
  4. The authors should delete figure Figure 2 since this figure is found by mistake twice in the manuscript and precisely before and after figure 1;
  5. About references: authors should substitute if they can following too old bibliographic references 44, 47, 48, 50, 88-90.

Author Response

  • All the changes done in blue color in the text

Reviewer 1

 The following acronym should be reported in exten so at their first appearence in the text: line 88 CVDs;

cardiovascular diseases

  • In the legend of figures and tables the authors should add informations about *P values and they should make it clear in the figures by placing asterisk/asterisks above each treatment group if differences with the reference group are significant;

We mentioned that the value of P≤0.05 and we add letters a, b, and c to differentiate between the significant.

During preparing the paper they move the letter away from the column

Fig. 6

Fig. 7

  • In addition statistical analysis should be revised;

Done

  • The authors should delete figure Figure 2 since this figure is found by mistake twice in the manuscript and precisely before and after figure 1;

This is mistake from the journal the original copy of submitted manuscript does not contain this error. We delete one of them

  • About references: authors should substitute if they can following too old bibliographic references 44, 47, 48, 50, 88-90.

All are method for determination of different parameters we cannot substitute.

We change 44  and 47 with the following

  1. Botsoglou, N. A.; Fletouris, D. J.; Papageorgiou, G. E.; Vassilopoulos, V. N.; Mantis, A. J. and Trakatellis, A. G.. Rapid, sensitive, and specific thiobarbituric acid method for measuring lipid peroxidation in animal tissue, food, and feedstuff samples. J. Agricult. Food Chem.,(1994), 42(9), 1931-1937.

47Jackson, S. D.; Halsall, H. B.; Pesce, A. J. and Heineman, W. R. Determination of serum alkaline phosphatase activity by electrochemical detection with flow injection analysis. Fresenius' J. Analy. Chem., 1993, 346(6), 859-862.

Reviewer 2 Report

The submitted work focused on evaluation of the effect of saffron extract on the hepatotoxicity induced by copper nanoparticles in mice is very interesting and could contribute a lot in the area of research. The manuscript is prepared precisely, design of experiment is planned and conducted well. Though, there are some shortcomings that must be improved and corrected.

First of all the entire text must be checked by English native speaker. Moreover, the meaning or explanations in some sentences must be improved. For example in lines 22-23: „This substance should also be useful and modern pharmacological methods for oxidative stress relief should be useful as new pharmacological tools for oxidative stress prevention.“ Or lines 617-619: „Thus, these substances should be useful as new pharmacological tools for alleviating oxidative stress should be useful as new pharmacological tools for alleviating oxidative stress.“

Some of abbreviations are not explained - in Keywords: line 24: TME, in Results: line 255: AST, ALT, ALP, line 266: LPO and under Table 2 (AST, ALP, ALT, ESE, CuNP).

Authors should specify saffron with latine name, family and genus in Abstract and when mentioned first time.

Sentences shouldn´t start with numbers, e.g. in lines 113, 202.

Change „◦C“ (line 112, 186) to „°C“.

Figure on the page 5 is the same as Figure 2.

Author Response

  • All the changes done in blue color in the text

Reviewer 2

The submitted work focused on evaluation of the effect of saffron extract on the hepatotoxicity induced by copper nanoparticles in mice is very interesting and could contribute a lot in the area of research. The manuscript is prepared precisely, design of experiment is planned and conducted well. Though, there are some shortcomings that must be improved and corrected.

First of all the entire text must be checked by English native speaker.

Done

Moreover, the meaning or explanations in some sentences must be improved. For example in lines 22-23: „This substance should also be useful and modern pharmacological methods for oxidative stress relief should be useful as new pharmacological tools for oxidative stress prevention.“

This substance should be useful as new pharmacological tools for oxidative stress prevention.

Or lines 617-619: „Thus, these substances should be useful as new pharmacological tools for alleviating oxidative stress should be useful as new pharmacological tools for alleviating oxidative stress.“

these substances should be useful as new pharmacological tools for alleviating oxidative stress.

Some of abbreviations are not explained - in Keywords: line 24: TME, in Results: line 255: AST, ALT, ALP, line 266: LPO and under Table 2 (AST, ALP, ALT, ESE, CuNP).

Keywords: TME should be TEM which is transmission electron microscope

Results: as aspartate aminotransferase (AST), alanine aminotransferase (ALT), and alkaline phosphatase (ALP)

LPO is lipid peroxidation

under Table 2 (AST, ALP, ALT, ESE, CuNP).

Done

Authors should specify saffron with latine name, family and genus in Abstract and when mentioned first time.

 Crocus sativus

We put it in line 71: Crocus sativus (saffron)

Sentences shouldn´t start with numbers, e.g. in lines 113, 202.

L-ascorbic acid aqueous solution (50 mL of 0.1M)

Thick ultrathin sections (60 nm) were

Change „◦C“ (line 112, 186) to „°C“.

Done

Figure on the page 5 is the same as Figure 2.

We remove one of them.